

# Machine learning can differentiate venom toxins from other proteins having non-toxic physiological functions

Ranko Gacesa[1], David J. Barlow[1] and Paul F. Long[1,2,3,4]

[1] Institute of Pharmaceutical Science, King's College London, London, United Kingdom
[2] Department of Chemistry, King's College London, London, United Kingdom
[3] Brazil Institute, King's College London, London, United Kingdom
[4] Faculdade de Ciências Farmacêuticas, Universidade de São Paulo, São Paulo, Brazil

## ABSTRACT

Ascribing function to sequence in the absence of biological data is an ongoing challenge in bioinformatics. Differentiating the toxins of venomous animals from homologues having other physiological functions is particularly problematic as there are no universally accepted methods by which to attribute toxin function using sequence data alone. Bioinformatics tools that do exist are difficult to implement for researchers with little bioinformatics training. Here we announce a machine learning tool called 'ToxClassifier' that enables simple and consistent discrimination of toxins from non-toxin sequences with >99% accuracy and compare it to commonly used toxin annotation methods. 'ToxClassifer' also reports the best-hit annotation allowing placement of a toxin into the most appropriate toxin protein family, or relates it to a non-toxic protein having the closest homology, giving enhanced curation of existing biological databases and new venomics projects. 'ToxClassifier' is available for free, either to download (https://github.com/rgacesa/ToxClassifier) or to use on a web-based server (http://bioserv7.bioinfo.pbf.hr/ToxClassifier/).

## INTRODUCTION

Falling costs of tandem mass spectrometry for shotgun proteomics have made generating vast amounts of protein sequence data increasingly affordable, yet the gap between obtaining these sequences and then assigning biological function to them continues to widen (*Bromberg et al., 2009*). Often, most sequences are deposited into protein databases with little, if any, accompanying experimental data from which biological functions can be inferred. Customarily, biological function is deduced indirectly by comparing amino acid sequence similarity to other proteins in large databases to calculate a ranking of proteins with respect to the query sequence. Using simple pair-wise comparisons as a sequence searching procedure, the BLAST suite of programs (for example, BLASTp) was first of its kind and has gained almost unprecedented acceptance among scientists (*Neumann, Kumar & Shalchian-Tabrizi, 2014*). Variations of the BLAST algorithm (for instance, PSI-BLAST (*Altschul et al., 1997*)) and development of probabilistic models (such as hidden

Corresponding author
Paul F. Long, paul.long@kcl.ac.uk

Markov models, HMMs (*Krogh et al., 1994*)) use multiple sequence alignments to detect conserved sequences (also referred to as motifs). Use of these models enables detection of remote homology between proteins seemingly unrelated when analysed by pairwise alignment alone (*Krogh et al., 1994*). Conversely, development of accurate algorithms and fast software tools that can automatically identify critical amino acid residues responsible for differences in protein function amongst sequences having exceptionally high sequence similarity remains a challenging problem for bioinformatics. In a post-genomic era, the toxins of venomous animals are an emerging paradigm.

Animal venoms comprise predominantly toxic peptides and proteins. Duplication of genes that formerly encoded peptides and proteins having non-toxic physiological functions is one of the foremost evolutionary drivers that gives rise to the enormous functional diversity seen in animal venom toxins (*Fry, 2005*; *Chang & Duda, 2012*). However, evidence is ambiguous as to whether these genes were expressed in multiple body tissues, with the duplicate copy then recruited into venom glands with subsequent neo-functionalisation to develop toxicity, or if there was duplication with ensuing sub-functionalisation of genes encoding pre-existing but non-toxic venom gland proteins (*Hargreaves et al., 2014*). Examples of both mechanisms have been demonstrated in different venomous animals (*Reyes-Velasco et al., 2015*; *Vonk et al., 2013*; *Junqueira de Azevedo et al., 2015*). Nonetheless, many toxin proteins that constitute venoms share a remarkable similarity to other proteins with non-toxic physiological functions, and deciphering sequence data to disentangle these functions is not a trivial task (*Kaas & Craik, 2015*).

Previous proteomic data from our laboratory and subsequent results of others realised an astonishingly high sequence similarity between cnidarian (jellyfish, coral, sea anemones etc.) toxins and those of other higher venomous animals (*Weston et al., 2012*; *Weston et al., 2013*; *Li et al., 2012*; *Li et al., 2014*). This suggested to us that a small number of sequences, when occurring in combination, might explain this similarity and prompted the search for toxin-specific motifs (*Starcevic & Long, 2013*). An unsupervised procedure was developed that resulted in the identification of motifs we called 'tox-bits', and which could describe most toxins as combinations of between 2–3 'tox-bits' (*Starcevic et al., 2015*). The 'tox-bits' are defined as HMM-profiles and were found to be superior at differentiating toxin from non-toxin sequences, when compared against standard BLAST or HMM based methods (*Starcevic et al., 2015*). However, implementation of 'tox-bits' HMM profiles is not straightforward for scientists with little or no bioinformatics experience. Hence, in this paper we introduce and make freely available an easy-to-use machine learning tool called the 'ToxClassifier' that employs 'tox-bits' HMM profiles and other standard classifier tools running in parallel to distinguish toxins from their non-toxic homologues.

## METHODS

### Datasets

Data for training and testing of machine learning classifiers used in ToxClassifier ensemble was obtained from UniProtKB database (*Bateman et al., 2015*), according to the following methodology:

(1) 'Positive' dataset representing well annotated putative animal toxins was downloaded from UniProtKB/SwissProt-ToxProt (*Jungo et al., 2012*). Database was searched for animal toxins and venoms, using search query: *taxonomy: "Metazoa [33208]" (keyword:toxin OR annotation:(type: "tissue specificity" venom)).* All duplicate entries with identical sequence or sequence identifier were removed, resulting in 8,093 sequences.

(2) 'Easy' negative dataset representative of physiological proteins was obtained by random sampling of 50,000 sequences in UniProtKB/SwissProt database (*Bateman et al., 2015*). All entries with duplicate sequence identifier or protein sequence were removed, as were all entries also occuring in *Positive dataset*; final dataset included 47,144 protein sequences.

(3) 'Moderate difficulty' negative dataset was designed to match highly curated toxin-like proteins with physiological function; it was created by BLASTp searching UniProtKB/SwissProt database with *Positive dataset*, with *e*-value cutoff of 1.0e−10. Resulting BLASTp hits were collected and all duplicates (with identical sequence or sequence identifier), and all sequences also present in *Positive dataset* or *Easy dataset* were removed, resulting in 8,034 proteins.

(4) 'Hard' negative dataset was constructed from TrEMBL database (*Bairoch & Apweiler, 2000*) instead of Swiss-Prot. As with *Moderate dataset*, it was created from results of BLASTp using *Positive dataset* as query and TrEMBL as target database. Duplicates and sequences also occurring in *Positive, Easy* or *Moderate datasets* were removed for total of 7,403 sequences.

All datasets are available for download at https://github.com/rgacesa/ToxClassifier/tree/master/datasets.

## Machine learning classifiers, training and testing

Models describing protein sequences were constructed as follows:

(1) *Single Amino acid frequency model* (OF): model uses length of sequence and frequency of each amino acid as input features.

(2) *Amino acid dimer frequency model* (BIF): model uses length of sequence, frequency of each amino acid and of each amino acid 2-mer.

(3) *Naive tox-bits model* (NTB): input features for this model are the number of 'tox-bits' for each 'tox-bits' HMM listed in the 'tox-bits' database (*Starcevic et al., 2015*).

(4) *Scored 'tox-bits' model* (STB): STB is a modification of the NTB model, with HMM bit-scores replacing the number of 'tox-bits' in each 'tox-bit' HMM model.

(5) *Tri-Blast Simple* (TBS) model: TBS uses BLASTp searches against positive (UniProtKB/SwissProt-ToxProt) and two negative control databases (close non-toxins from UniProtKB/SwissProt and non-toxins from TrEMBL); features include *bit-score, query length, subject length, query/subject length ratio, query coverage, percentage of identity, percentage of positive matches;* features also include amino-acid frequencies. Scores are computed from the 'best-hit' in each database, with a BLAST *e*-value of 1.0e−10.

(6) *Tri-Blast Enhanced A* (TBEa) model: TBEa model is an expanded variant of TBS, with amino dimer frequencies included in the model.

(7) *Tri-Blast Enhanced B* (TBEb): model is a variation of TBEa, trained on 80% of the input dataset and with a BLAST *e*-value cut-off value of 1.0e+3 for the detection of similar toxin or non-toxic sequences.

Feature extraction and vectorisation was implemented using the Python 2.7 (https://www.python.org/download/releases/2.7/) scripting language, NCBI BLAST+ version 2.2.31 (*Camacho et al., 2009*), HMMER 3.1b1 (*Eddy, 2011*), the 'tox-bit' HMM profile database (*Starcevic et al., 2015*) and a set of custom BLAST databases based on UniProtKB/SwissProt-ToxProt, UniProtKB/SwissProt and TrEMBL databases. Vectorization scripts, vectorised data sets and trained models are available for download at https://github.com/rgacesa/ToxClassifier.

Support Vector Machine (SVM), Gradient Boosted Machine (GBM) and Generalised Linear Model (GLM) classifiers were trained for each of the models; Classifiers were implemented using the R-programming language, Caret package (http://topepo.github.io/caret/index.html). The training set for each dataset was selected by random sampling of 75% of the sequences, and combined training sets were used to train the classifiers. Input features were 0-centered and scaled by standard deviation and training was performed with 10 internal bootstraps. Learning curves were constructed for each classifier to evaluate training efficiency and potential overtraining by plotting performance versus number of sequences in training set.

Classifiers were tested using those sequences from *Positive, Easy, Moderate* and *Hard* datasets not used in training. Performance was evaluated on each of the datasets and on the summary dataset. The following performance measurements were calculated:

(1) *Number of true positives* (TP): number of toxic sequences in dataset correctly predicted as toxins; sequence was considered a toxin if listed in UniProtKB/ToxProt database.

(2) *Number of true negative*s (TN): number of non-toxic sequences in dataset which are correctly predicted as non-toxic (not in UniProtKB/ToxProt database).

(3) *Number of false positive* (FP): number of non-toxic sequences in dataset incorrectly classified as toxins.

(4) *Number of false negatives* (FN): number of toxic sequences in dataset incorrectly classified as non-toxic.

(5) *Accuracy* (ACC): accuracy is calculated as proportion of sequences correctly classified as toxins or non-toxins; $ACC = (TP + TN) / (TP + TN + FP + FN)$.

(6) *Specificity* (SPEC): also called *true negative rate*, specificity is proportion of correctly predicted non-toxins (true negatives); $SPEC = TN / (TN + FP)$.

(7) *Sensitivity* (SENS): also called *true positive rate* or *recall*, sensitivity is proportion of correctly predicted toxins (true positives); $SENS = TP/(TP + FN)$.

(8) *Balanced accuracy* (B.ACC): balanced accuracy is a mean value of specificity and sensitivity; $BACC = (SPEC + SENS)/2$.

(9) *Negative predictive value* (NPV): proportion of negatives that are true negatives. $NPV = TN/(TN + FN)$.

(10) *Positive predictive value* (PPV): also called Precision, PPV measures proportion of positives which are true positives; $PPV = TP/(TP + FP)$.

(11) *F-Score* (F1): F-score is harmonic mean of precision and sensitivity and represents weighted average of precision and recall. It is calculated as $F1 = 2 * TP/(2 * TP + FP + FN)$.

(12) Matthews' correlation coefficient (MCC): the MCC value (also known as phi-coefficient) is a measure of correlation between observed and predicted. It is considered balanced measure even for classes with different amounts of positive and negative values; $MCC = \frac{TP*TN - FP*FN}{\sqrt{(TP+FP)*(TP+FN)*(TN+FP)*(TN+FN)}}$ (*Matthews, 1975*; *Powers, 2011*).

## Conventional annotation models

Annotation models simulating manual annotation were constructed based on BLAST and HMMER, as follows:

(1) Naive-BLAST: annotation method is based on the assumption that a sequence is a toxin if a BLAST search against the UniProtKB/SwissProt-ToxProt dataset returns a positive hit at a certain *e*-value (usually between 1.0e–20 and 1.0e–5). This approach and its variations are commonly encountered in the literature (*Sher & Zlotkin, 2009*; *Schwartz et al., 2015*; *Liu et al., 2015*; *Whittington et al., 2010*).

(2) OneBLAST: this approach classifies a sequence by a single BLAST search against databases including UniProtKB/SwissProt-ToxProt and other 'toxin-like' but non-toxic sequences extracted from UniProtKB/SwissProt and TrEMBL databases (combination of *Positive, Moderate* and *Hard* datasets). A sequence is classified as a putative toxin if the highest scoring BLAST hit at a selected *e*-value is from UniProtKB/SwissProt-ToxProt; it is classified as non-toxic if top BLAST hit is not from UniProtKB/SwissProt-ToxProt, or if all BLAST hits are above the selected *e*-value.

(3) TriBLAST: annotation is performed by a variation of OneBLAST, using separate BLAST searches against UniProtKB/SwissProt-ToxProt and two non-toxin databases (for the *Moderate* and *Hard* datasets); a sequence is classified as a toxin if the BLAST search against UniProtKB/SwissProt-ToxProt returns a hit below a certain *e*-value and with a higher score than the searches against the other databases. Variations of this method are commonly used in toxin annotation (*Gacesa et al., 2015*; *Rachamim et al., 2014*).

(4) hmmerToxBits: this method is a variation of the naive-BLAST, using HMMER package Hmmsearch instead of BLAST, and the database of 'tox-bits' HMM models as the target database. A sequence is classified as a toxin if one or more HMMs can be detected within a certain *e*-value cut-off.

(5) hmmerVenom: modification of hmmerToxBits, the method uses HMM profiles extracted by 'venom' and 'toxin' text search of the Pfam database instead of 'tox-bit' HMMs.

(6) twinHmmerPfam: a HMMER based variant of TriBLAST approach, this method performs hmmsearches against two HMM databases (toxin/positive HMMs and negative control) and compares bitscores. Sequence is annotated as a toxin if bitscore for toxin HMM database is higher. TwinHmmerPfam toxin HMMs were extracted from Pfam by keyword search for 'toxin' and 'venom,' while negative control database is comprised from remainder of Pfam.

(7) twinHmmerToxBits: a variation of twinHmmerPfam method, method compares hmmsearch against toxin HMMs and negative control. For this model, positive database is composed from 'tox-bit' HMMs derived from Tox-Prot toxins (*Starcevic et al., 2015*), while negative control is Pfam database from which HMMs containing 'toxin' or 'venom' keywords were removed.

BLAST databases for Naive-BLAST, OneBLAST and TriBLAST were constructed from 75% randomly sampled sequences in *Positive, Easy, Moderate* and *Hard* and performance was measured by annotating remainder of data. HMMER based methods were tested with 25% of random sequences from input datasets, using all appropriate HMM models. Database construction and testing was repeated 10 times and results were averaged.

## ToxClassifier Meta-classifier calibration and testing

ToxClassifier meta-classifier was constructed from nine annotation model and classifier combinations (BIF_SVM, BIF_GBM, STB_SVM, STB_GBM, TBS_SVM, TBS_SVM, TBEa_SVM, TBEb_SVM and TBEb_GBM). Each of nine classifiers reports 1 if the input sequence is predicted as toxin or 0 if predicted as non-toxic, for final prediction score of 0 to 9. Datasets used for calibration of meta-classifier were chosen from the set of venomous and non-venomous animals (human *Homo sapiens*, the house mouse *Mus musculus*, the Burmese python *Python bivittatus*, king cobra *Ophiophagus hannah*, the duck-billed platypus *Ornithorhynchus anatinus*, the snakelocks sea anemone *Anemonia viridis*, the starlet sea anemone *Nematostella vectensis*; and all proteins deposited in the UniProtKB/SwissProt and TrEMBL databases attributed to snakes, spiders, wasps and *Conus* snails). All sequences were downloaded from UniProtKB database with exception of *Python bivittatus* which was not available in UniProtKB and was downloaded from NCBI protein database (*Wheeler et al., 2003*); all data is available at https://github.com/rgacesa/ToxClassifier/tree/master/datasets. Datasets were split into training sets consisting of 75% of data and test sets including remaining 25% of sequences.

ToxClassifier meta-classifier was calibrated by evaluating prediction score versus performance for each animal training set and for summary dataset constructed by combining animal training datasets with exclusion of *Conus snail* data, which was dropped due to suspected low quality of annotation. Calibrated ToxClassifier, with prediction score 5 or more as cut-off for positive classification, was tested on animal data test sets, and performance measures were compared to OneBLAST, NaiveBLAST models and ClanTox server. Data used for training and testing and all calculated performance metrics are available at https://github.com/rgacesa/ToxClassifier/tree/master/datasets/toxclassifier_calibration_test.

## ToxClassifier comparison to other published tools

Performance of ClanTox server (*Kaplan, Morpurgo & Linial, 2007*) was tested on *Positive, Easy, Moderate* and *Hard* datasets and on *animal data test sets*. ToxinPred server (*Gupta et al., 2013*) was tested on 868 *Positive dataset* sequences of length up to 30 amino acids (ToxinPred sequence length limit) and on negative dataset composed 30 amino acid or shorter protein sequences randomly selected from UniProt database (5,673 non-duplicate, non-ToxProt sequences). ToxinPred was not tested on animal data due to lack of short

sequences available in these datasets. SpiderP server (*Wong et al., 2013*) was not tested as service was not available at the time and PredCSF server (*Fan et al., 2016*) was not tested as it was deemed too specialized and only accepts single sequence as input.

### User interface

The ToxClassifier web service front-end is implemented using HTML 5.1 (https://www.w3.org/html/), JavaScript (https://www.javascript.com/), jQuery (https://jquery.com/), CSS (https://www.w3.org/Style/CSS/), Java 7.0 (https://www.oracle.com/java/index.html) and the Java Server Pages (JSP 2.1) framework (http://www.oracle.com/technetwork/java/javaee/jsp/index.html). Visualisation is performed using R (https://www.r-project.org/), ggplot2 (http://ggplot2.org/) and R markdown (http://rmarkdown.rstudio.com/) packages. ToxClassifier runs on an Apache Tomcat 8.0 web server (http://tomcat.apache.org/download-80.cgi), under an Ubuntu Linux 12.04 operating system (http://www.ubuntu.com/). The service is hosted by the Section of Bioinformatics, Faculty of Food Technology and Biotechnology, University of Zagreb, Croatia (http://www.pbf.unizg.hr/en/departments/department_of_biochemical_engineering/section_for_bioinformatics).

## RESULTS

The accuracy of three individual machine-learning classifiers to predict toxins from proteins having other physiological functions was assessed by training each classifier using seven different annotation models. The learning classifiers were a Support Vector Machine (SVM) and Gradient Boosted Machine (GBM) chosen as high-performing predictors, and a Generalised Linear Model (GLM) regarded as a simple classifier, but with which a baseline could be established that would allow comparison of the performance of the SVM and GBM machines. A detailed description of the annotation models is given in 'Methods' section, but briefly the annotation models used the following sequence information from the training set as classifier inputs: either the frequency of amino acids (TBSim) or combinations of two amino-acids (BIF), the presence of absence or 'tox-bits' (SToxA), HMM scores for 'tox-bits' (SToxB), a selection of BLAST output co-variants (TBEa) or a variation on TBSim and TBEa (TBEb).

The training set was constructed by merging 75% of arbitrarily selected sequences from each of the following datasets: 1/ A 'Positive' dataset that contained all 8,093 protein sequences deposited in the UniProtKB/SwissProt-ToxProt database of animal venom toxins (*Jungo et al., 2012*). 2/ An 'Easy' dataset composed of 47,144 random, non-duplicate sequences from UniProtKB/SwissProt database (*Bateman et al., 2015*). 3/ A 'Moderate' dataset comprised of 8,034 unique sequences from the manually curated UniProtKB/SwissProt database considered to be non-toxic but with homology to toxin proteins in UniProtKB/SwissProt-ToxProt. 4/ A 'Hard' dataset that included 7,403 non-duplicate sequences extracted from the computer annotated TrEMBL (*Bairoch & Apweiler, 2000*) database, also considered to be non-toxic but with homology to animal venom toxins in UniProtKB/SwissProt-ToxProt.

All training was performed using 10 internal bootstrap cross-validations on the training set, and learning curves showing the accuracy of predictions versus the number of sequences

in the training sets were constructed, thereby allowing a comparative evaluation of training efficiency (Fig. S1). The trained classifiers were then tested for prediction accuracy using the remaining 25% of sequences not included in the training set. Performance values were calculated to give an overall comparative classification of protein sequences as toxins or non-toxins (Table 1). By comparing learning curves (Fig. S1) and accuracy of predictions (Table 1), 9 of the annotation model and classifier combinations were chosen to construct the 'ToxClassifier' ensemble. The trained classifiers were: BIF_SVM, BIF_GBM, STB_SVM, STB_GBM, TBS_SVM, TBS_SVM, TBEa_SVM, TBEb_SVM and TBEb_GBM. These classifiers all gave excellent accuracy scores to predict toxins from the Positive dataset (range 0.82–0.96) and non-toxin proteins from the Easy, Moderate and Hard datasets (range 0.92–1.00). No GLM classifiers were included in the ensemble because prediction accuracies were considerably lower when compared with SVM and GBM machines. Classifiers using NTB and OF annotation models were also abandoned in favour of better performing STB and BIF models. Furthermore, the TBEa_GBM model consistently underperformed compared to the TBE_SVM model and was excluded, giving an odd number of classifiers in the ensemble, thereby avoiding a 'tied vote' scenario when the outputs were interpreted collectively.

The prediction accuracy of the trained machine learning classifiers was next compared to more conventional annotation methods based on sequence alignment, to determine if machine learning predictions were superior or inferior to well established and accepted bioinformatics tools. A detailed description of the annotation models based on these bioinformatics tools is given in 'Methods.' Briefly, simple predictions were made by taking the best-hit from BLAST comparisons between a query sequence and the UniProtKB/SwissProt-ToxProt database (naiveBLAST method), or the best-hit following a HMMER hmmsearch comparison between a query sequence and either existing HMM models for toxin protein families in the Pfam database (hmmerVenom model), or our own 'tox-bits' HMM models (hmmerToxbits classifier). More sophisticated annotation models also used BLAST or HMMER searches, but the best-hit was extracted following simultaneous comparisons between the query sequence and multiple datasets. These sophisticated annotation models are also described in 'Methods', but briefly these models were constructed from sequence information extracted from either UniProtKB/SwissProt-ToxProt sequences supplemented with additional toxin-like sequences from the UniProtKB/SwissProt and TrEMBL databases, or UniProtKB/SwissProt-ToxProt sequences supplemented with non-toxin sequences from the 'Moderate' and 'Hard' datasets used to train the machine classifiers. Training and test sets were analogous in design and execution to the machine classifier learning, with 75% of sequence information used to construct the BLAST and HMMER databases and the remaining 25% of data used to evaluate performance. Prediction accuracy measures for each query sequence using each of the bioinformatics models were repeated 10 times to give a final balanced accuracy value. Accuracy measure calculations are described in 'Methods'. A range of sequence-alignment scoring was also tested to select the lowest BLAST and HMMER cut-off scores that gave the most precise toxin annotation. This value was 1.0e-20 for both BLAST and HMMER searches (Fig. S2).

**Table 1** **Prediction accuracy on positive and negative datasets, as well as range of measurements calculated for all test data, and described in detail in 'Methods.'** Annotation models used as classifier inputs either: the frequency of amino acids (TBSim) or combinations of two amino-acids (BIF); the presence of absence or 'Tox-Bits' (SToxA); HMM scores for 'ToxBits' (SToxB); a selection of BLAST output co-variants (TBEa); a variation on TBSim and TBEa (TBEb). Classifier Learning Machines used were: Gradient Boosted (GBM), Support Vector (SVM) and Generalised Linear Model (GLM). The datasets were a 'Positive' control containing only validated animal toxins, an 'Easy' dataset composed of non-toxin sequences, a 'Moderate' dataset comprising curated non-toxin sequences but with homology to 'Positive' sequences, and a 'Hard' dataset that included all sequences from the 'Moderate' dataset, together with un-curated sequences also with homology to 'Positive' sequences.

| Annotation model | Classifier | Classification scores for: | | | | Test set summary | | | | | |
|---|---|---|---|---|---|---|---|---|---|---|---|
| | | Accuracy (Positive toxin dataset) | Accuracy (Easy non-toxin dataset) | Accuracy (Moderate non-toxin dataset) | Accuracy (Hard non-toxin dataset) | PPV | NPV | Sens. | Spec. | F1-value | MCC |
| TBSim | GBM | 0.80 | 0.99 | 0.98 | 0.92 | 0.80 | 0.98 | 0.84 | 0.97 | 0.82 | 0.80 |
| | SVM | 0.80 | 1.00 | 0.98 | 0.94 | 0.80 | 0.99 | 0.91 | 0.97 | 0.85 | 0.84 |
| | GLM | 0.55 | 0.99 | 0.96 | 0.84 | 0.55 | 0.97 | 0.69 | 0.94 | 0.61 | 0.57 |
| BIF | GBM | 0.83 | 1.00 | 0.98 | 0.94 | 0.83 | 0.99 | 0.92 | 0.98 | 0.87 | 0.86 |
| | SVM | 0.89 | 1.00 | 0.98 | 0.96 | 0.89 | 0.99 | 0.94 | 0.99 | 0.91 | 0.90 |
| | GLM | 0.71 | 0.99 | 0.99 | 0.91 | 0.71 | 0.98 | 0.82 | 0.96 | 0.76 | 0.74 |
| SToxA | GVM | 0.64 | 1.00 | 0.98 | 0.94 | 0.64 | 0.99 | 0.90 | 0.96 | 0.75 | 0.73 |
| | SVM | 0.84 | 1.00 | 0.96 | 0.91 | 0.84 | 0.98 | 0.87 | 0.98 | 0.86 | 0.84 |
| SToxB | GBM | 0.75 | 1.00 | 1.00 | 0.93 | 0.75 | 0.99 | 0.92 | 0.97 | 0.83 | 0.84 |
| | SVM | 0.85 | 1.00 | 0.99 | 0.92 | 0.85 | 0.99 | 0.91 | 0.98 | 0.88 | 0.81 |
| | GLM | 0.03 | 1.00 | 0.99 | 0.99 | 0.03 | 1.00 | 0.61 | 0.89 | 0.06 | 0.12 |
| TBEa | GBM | 0.88 | 1.00 | 1.00 | 0.99 | 0.88 | 1.00 | 0.99 | 0.98 | 0.93 | 0.93 |
| | SVM | 0.93 | 1.00 | 1.00 | 0.97 | 0.93 | 1.00 | 0.97 | 0.99 | 0.95 | 0.94 |
| | GLM | 0.96 | 1.00 | 1.00 | 0.94 | 0.96 | 0.99 | 0.95 | 0.99 | 0.95 | 0.95 |
| TBEb | GBM | 0.82 | 1.00 | 1.00 | 1.00 | 0.82 | 1.00 | 1.00 | 0.98 | 0.90 | 0.90 |
| | SVM | 0.96 | 1.00 | 1.00 | 0.97 | 0.96 | 1.00 | 0.97 | 0.99 | 0.97 | 0.96 |
| | GLM | 0.93 | 1.00 | 1.00 | 0.99 | 0.93 | 1.00 | 0.99 | 0.99 | 0.96 | 0.95 |

Machine learning classifiers were also evaluated against currently available published tools for toxin prediction and annotation; Animal toxin prediction server ClanTox (*Kaplan, Morpurgo & Linial, 2007*) was benchmarked using *Positive, East, Moderate* and *Hard* datasets and summary of these datasets. As ToxinPred (*Gupta et al., 2013*) tools predict only small peptide toxins, it was tested using a subset of *Positive dataset* with sequences no longer than 30 amino acids (868 sequences) and separate negative dataset composed of 5,673 random short proteins from UniProtKB database. SpiderP (*Wong et al., 2013*) was not benchmarked as the server no longer seems publically available. Finally, PredCSF (*Fan et al., 2016*) is a conotoxin-specific tool and was deemed not comparable to general annotation tools; it also only allows single sequence input, making it unsuitable for large scale testing. Final performance measures compared between the different tools are listed in Table 2.

Testing of sequence-alignment based annotation models (Table 2) demonstrated that the simplistic methods (naiveBLAST, hmmerToxBits and hmmerVenom) gave high prediction accuracies for sequences in the Easy dataset (ranging from 0.95 for hmmerVenom to 0.99 for naiveBLAST), but underperformed in annotation of the physiological toxin-like sequences in the Moderate and Hard datasets (accuracies ranging from 0.74 to 0.83 for Moderate and 0.07 to 0.23 for Hard dataset (the poor performance here, also evinced by the low F1 and MCC scores)). More sophisticated BLAST-based methods (oneBLAST and triBLAST) gave very high prediction accuracy scores (0.93–0.999) for sequences in the Easy and Moderate datasets, but somewhat lower performance on sequences in the Positive and Hard datasets (0.86–0.90). Pfam-based twinHMMER gave the highest accuracy prediction for non-toxin sequences, but underperformed compared to the other annotation models against sequences in the positive toxin dataset (accuracy 0.56). The 'tox-bits' based variant accurately predicted sequences in the Easy and Moderate datasets (accuracy 0.85–0.999), but suffered from a high false positive rate when sequences in the Hard dataset were analysed (accuracy 0.44). When compared to machine learning-based methods, even the most accurate of the sequence alignment-based models (oneBLAST and triBLAST) were surpassed by the majority of the machine learning based classifiers, especially by TBEa and TBEb models (SVM and GBM variants), which gave the highest accuracy of prediction for sequences in all test datasets. All prediction methods showed higher performance for negative prediction (predicting non-toxin as non-toxin) compared to positive prediction (correctly predicting toxin as toxic), with Specificity (Spec) and Negative Prediction Value (NPV) significantly higher than Sensitivity (Sens) and Positive Prediction Value (PPV).

Each of the 9 machine learning classifiers used in the 'ToxClassifier' ensemble gives a simple bit (1 or 0) value as output to predict whether the likely biological activity of the input sequence is as a toxin (1), or has a non-toxic (0) physiological role and scores are summed into final prediction score ranging from 0 to 9. Evaluation of this final prediction score was performed on test sets obtained from randomly sampling 75% of sequences in the published annotated genomes from a selection of venomous animals (king cobra *Ophiophagus hannah*, the duck-billed platypus *Ornithorhynchus anatinus*, the snakelocks sea anemone *Anemonia viridis*, the starlet sea anemone *Nematostella vectensis*; and all proteins deposited in the UniProtKB/SwissProt and TrEMBL databases attributed to snakes,

Gacesa et al. (2016), *PeerJ Comput. Sci.*, DOI 10.7717/peerj-cs.90

**Table 2 Performance for selected annotation models and published toxin prediction tools.** Prediction accuracy is listed for positive and negative datasets and measurements are also shown for summary of all test data. ToxinPred server, marked with star and displayed in italic, was tested with short protein sequences only. Annotation models were constructed from sequence information extracted from either: BLAST (naiveBLAST), 'tox-bits' HMM (hmmerToxBits) or Pfam HMM (hmmerVenom) comparisons with the UniProtKB/SwissProt-ToxProt database; or BLAST (oneBLAST), 'tox-bits' HMM (twinHmmerPfam) or Pfam HMM (twinHmmerPfam) comparisons with the UniProtKB/SwissProt-ToxProt sequences supplemented with additional toxin-like sequences from the UniProtKB/SwissProt and TrEMBL databases; or BLAST (triBLAST) comparisons with the UniProtKB/SwissProt-ToxProt sequences supplemented with non-toxin sequences from the 'Moderate' and 'Hard' datasets used to train the machine classifiers. The datasets were a 'Positive' control containing only validated animal toxins, an 'Easy' dataset composed of non-toxin sequences, a 'Moderate' dataset comprising curated non-toxin sequences but with homology to 'Positive' sequences, and a 'Hard' dataset that included all sequences from the 'Moderate' dataset, together with un-curated sequences also with homology to 'Positive' sequences.

| Annotation model | Tool | Classification scores for: | | | | Test set summary | | | | | |
|---|---|---|---|---|---|---|---|---|---|---|---|
| | | Accuracy (Positive toxin dataset) | Accuracy (Easy non-toxin dataset) | Accuracy (Moderate non-toxin dataset) | Accuracy (Hard non-toxin dataset) | PPV | NPV | Sens. | Spec. | F1-value | MCC |
| naiveBLAST | BLAST | 0.90 | 0.99 | 0.83 | 0.07 | 0.90 | 0.86 | 0.46 | 0.99 | 0.60 | 0.58 |
| oneBLAST | BLAST | 0.86 | 1.00 | 0.98 | 0.90 | 0.86 | 0.99 | 0.89 | 0.98 | 0.87 | 0.86 |
| triBLAST | BLAST | 0.87 | 0.99 | 0.93 | 0.87 | 0.87 | 0.97 | 0.78 | 0.98 | 0.82 | 0.80 |
| hmmerToxBits | HMMER | 0.91 | 0.99 | 0.80 | 0.19 | 0.91 | 0.87 | 0.48 | 0.99 | 0.63 | 0.60 |
| hmmerVenom | HMMER | 0.65 | 0.95 | 0.74 | 0.23 | 0.65 | 0.84 | 0.34 | 0.95 | 0.45 | 0.38 |
| twinHmmerPfam | HMMER | 0.56 | 0.99 | 0.98 | 0.91 | 0.56 | 0.98 | 0.78 | 0.95 | 0.65 | 0.62 |
| twinHmmerToxBits | HMMER | 0.85 | 1.00 | 0.93 | 0.44 | 0.85 | 0.92 | 0.59 | 0.98 | 0.70 | 0.67 |
| ClanTox server | ML | 0.66 | 0.99 | 0.93 | 0.73 | 0.66 | 0.95 | 0.65 | 0.96 | 0.66 | 0.61 |
| *ToxinPred server** | *ML* | *0.55* | *0.98* | *N/A* | *N/A* | *0.55* | *0.98* | *0.82* | *0.93* | *0.66* | *0.63* |

spiders, wasps and *Conus* snails) and other animals considered to be non-venomous (human *Homo sapiens*, the house mouse *Mus musculus* and the Burmese python *Python bivittatus*). Calibration was performed by assessing the performance measures of the Toxclassifier ensemble relative to prediction score; calibration curves for summary of all animal genome data are presented in Fig. 1. When the average correct annotation of all input sequences for all genomes was calculated, a combined score from five out of the nine classifiers giving correct classification provided a good balance between the detection of toxins and the filtering of non-toxins. Hence, a calibration for the ToxClassifier ensemble was possible where an input sequence giving a combined score of >6 would be considered a likely toxin, a combined score of <3 would be regarded as non-toxic, while an input sequence presenting with a score 4 or 5 would suggest a potential toxin, but would require manual evaluation using additional tools, for example, InterProScan (*Zdobnov & Apweiler, 2001*).

Performance of calibrated 'ToxClassifier' meta-classifier was evaluated on a test set comprising 25% of the animal genome data not used for calibration; these results were compared to naiveBLAST and OneBLAST conventional methods and to ClanTox server for animal toxin prediction (*Kaplan, Morpurgo & Linial, 2007*). Performance measurements are reported in Table 3 and comparison of F1-scores and MCC values across all datasets is presented in Fig. 2; Figs. S3/A and S3/B. Finally, the 'ToxClassifier' was assessed in a blinded experiment that used as input a set of protein sequences derived from the venom gland transcriptome of the Amazonian rain forest pit viper *Bothrops atrox* (Data S1). The sequences had been annotated using standard methods and manually inspected, with the biological activities of some also being authenticated experimentally. The results of the 'ToxClassifier' predictions matched with the expert annotation (Table 4).

## DISCUSSION

The continued decline in proteomics sequencing costs over recent years has led to an explosion in venomics data characterising the toxic peptide and protein components in many venomous animals (*Kaas & Craik, 2015*). However, there is currently no widely accepted and standard method for functional annotation of toxins from these data sources, leading to inconsistent estimates for the number of toxins in the venom of the same animal. For example, the venom of the duck-billed platypus *Ornithorhynchus anatinus* has only 6 toxins listed following manual annotation in the latest release of the UniProtKB/SwissProt-ToxProt database (11th May 2016), yet 107 putative toxins were identified by a simple pair-wise BLASTp search using venom gland transcriptome sequences as input to search the UniProtKB/SwissProt ToxProt database (*Jungo et al., 2012*). In addition to separate homology searching methods to interpret the same data, many venomics projects now also include different manual filtering steps as part of the annotation process (*Rachamim et al., 2014*; *Gacesa et al., 2015*), exacerbating the problem of results verification.

In this study, a selection of machine learning-based classifiers implementing a range of BLAST and HMMER-based annotation models were trained on datasets of known toxins, protein sequences assumed to be non-toxic but with homology to known toxins, and predicted proteins encoded in the genome, transcriptome or proteome of a range of

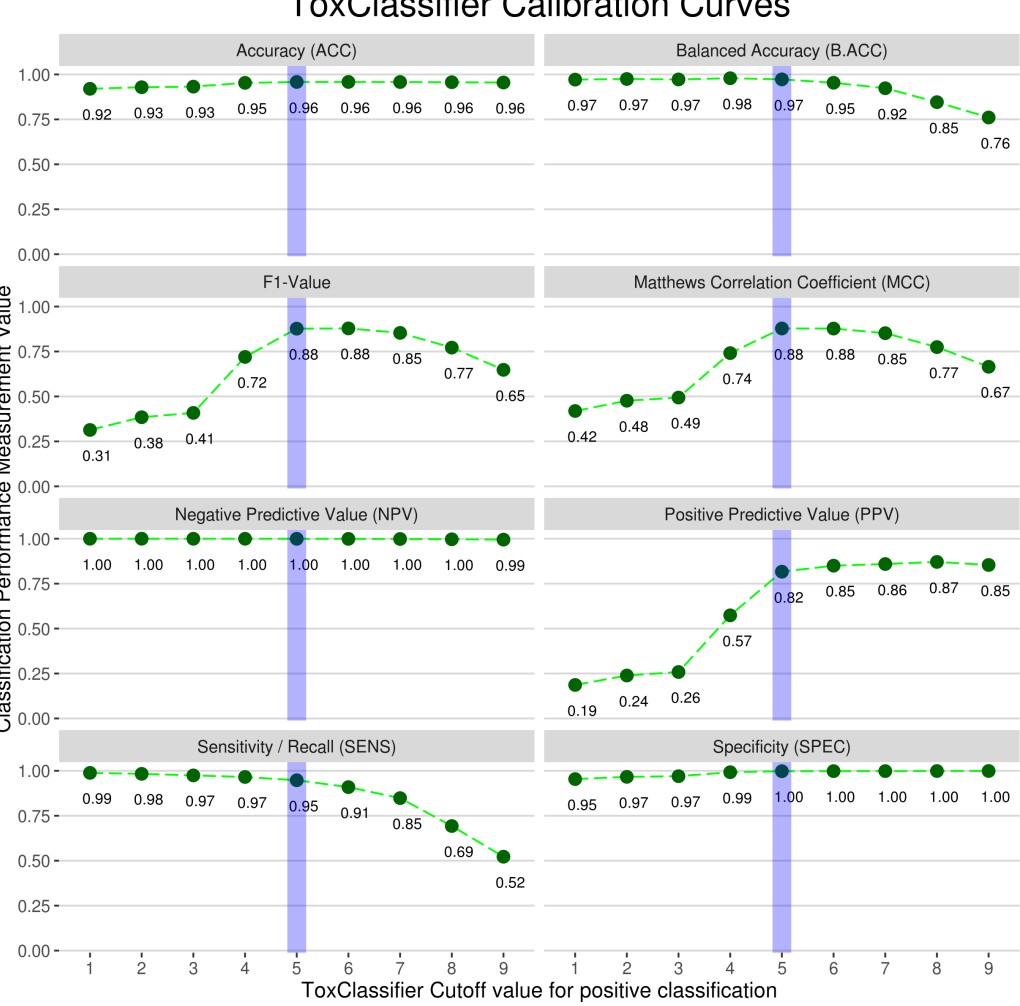

## ToxClassifier Calibration Curves

**Figure 1** **Calibration curves used to select final prediction scores for ToxClassifier ensemble.** Each of performance measures, described in detail in Methods, is shown for ToxClassifier prediction with toxin prediction cut-off values 1–9. Dotted green line shows trends of prediction measurement with increase in ToxClassifier cut-off value and final cut-off value implemented in calibrated ToxClassifier is highlighted with blue line.

venomous and non-venomous animals. A comparison between the results presented in Tables 1–3 demonstrated that the majority of the machine learning methods consistently out-performed standard bioinformatics approaches of functional annotation. Interestingly, all tested methods demonstrated higher performance for negative prediction (classification of non-toxic sequences) compared to positive classification (prediction of toxic sequences as toxins). These results demonstrate that differentiating between physiological toxin-like proteins and actual toxins is more difficult then prediction of random proteins as non-toxic, which is to be expected considering the similarity and common origin of many toxins and toxin-like sequences (*Fry, 2005*; *Chang & Duda, 2012*; *Hargreaves et al., 2014*). As such, it is important to consider balanced performance measurements when assessing toxin classifiers, with scores such as F1-score and MCC value (*Matthews, 1975*; *Powers, 2011*) providing

**Table 3  ToxClassifier calibrated meta-classifier, Test set results compared to oneBLAST, naiveBLAST and ClanTox annotation methods.** Table lists comparison of classification performance for calibrated ToxClassifier to BLAST based annotation models and ClanTox toxin prediction server. All tests were conducted test dataset not used in calibration of ToxClassifier.

| Classification performance of ToxClassifier meta-classifier, compared to BLAST based methods and ClanTox server | | | | | | | | |
|---|---|---|---|---|---|---|---|---|
| Annotation model | Tool | Accuracy | Positive prediction value | Negative prediction value | Sensitivity | Specificity | F1-value | MCC value |
| naiveBLAST | BLAST | 0.976 | 0.294 | 0.998 | 0.857 | 0.977 | 0.438 | 0.494 |
| oneBLAST | BLAST | 0.994 | 0.660 | 1.000 | 0.951 | 0.995 | 0.779 | 0.790 |
| ClanTox server | ML-based | 0.947 | 0.209 | 1.000 | 0.954 | 0.976 | 0.342 | 0.440 |
| ToxClassifier | ML-based | 0.997 | 0.825 | 1.000 | 0.967 | 0.998 | 0.890 | 0.892 |

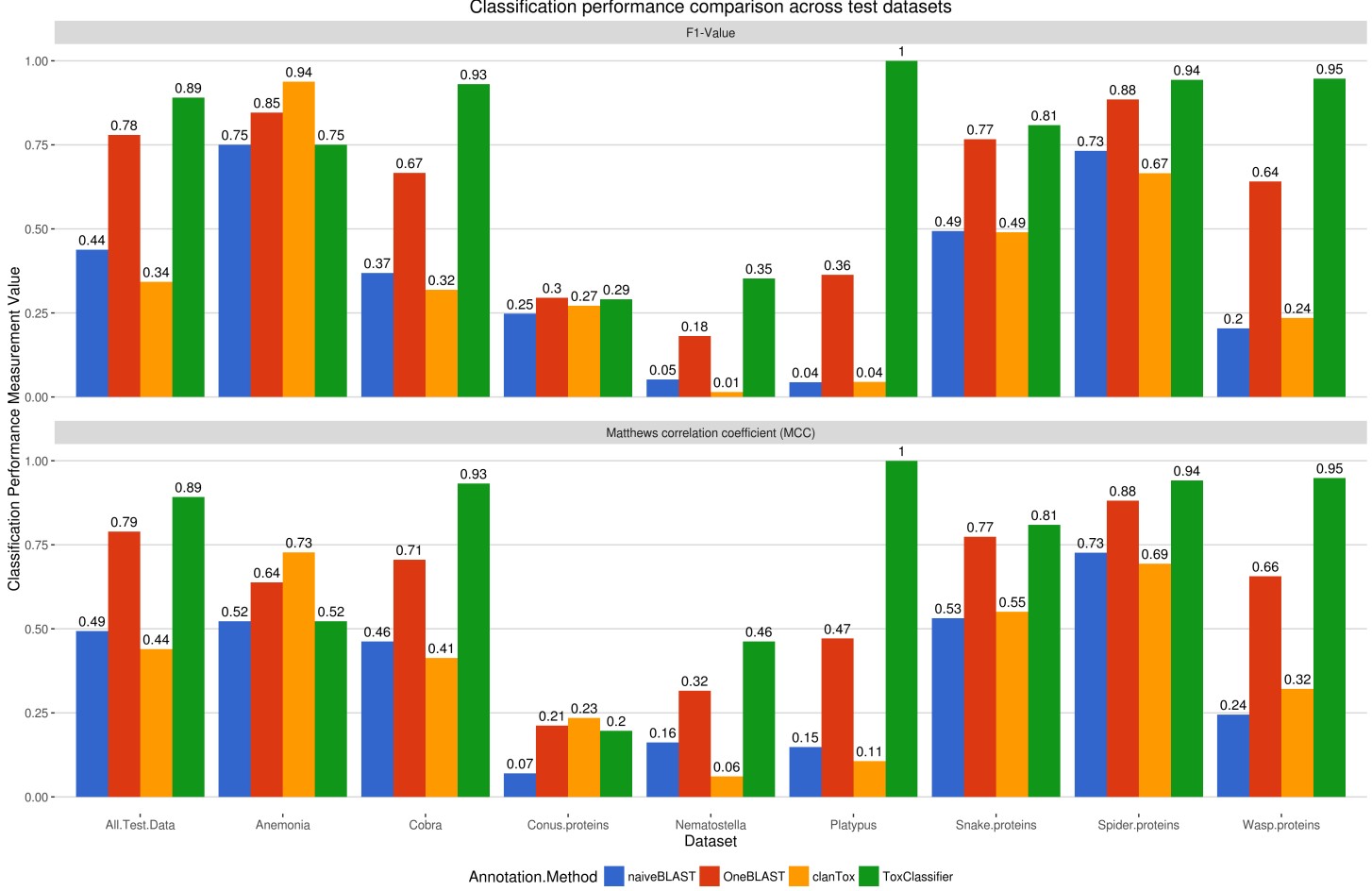

**Figure 2  Comparison between selected toxin prediction tools for all animal test datasets.** Calibrated ToxClassifier, with positive prediction cutoff 5, is shown by green bar, ClanTox prediction server is displayed in orange, oneBLAST prediction performance in red and naiveBLAST in blue. F1-value and Matthews' correlation coefficient are displayed for each of test sets and for summary of all data with exclusion of conus snail proteins.

**Table 4  Results of expert manual annotation to ToxClassifier annotation for set of novel proteins from venom gland transcriptome of _Bothrops atrox_ snake.**

| | Sequence annotation | |
|---|---|---|
| Test sample | 'ToxClassifier' | Standard annotation |
| Test 01 | Potential toxin | BPP precursor—Toxin |
| Test 02 | Potential toxin | Crisp—Toxin |
| Test 03 | Toxin | CTL—Toxin |
| Test 04 | Toxin | CTL—Toxin |
| Test 05 | Potential toxin | Hyaluronidase—Toxin |
| Test 06 | Potential toxin | Kunitz –like sequence—Probably a toxin |
| Test 07 | Toxin | LAAO—Toxin |
| Test 08 | Not a toxin | Nucleotidase—Probably not a toxin |
| Test 09 | Not a toxin | Phosphodiesterase— Probably not a toxin |
| Test 10 | Toxin | PLA2—Toxin |
| Test 11 | Toxin | SVMP class PI—Toxin |
| Test 12 | Toxin | SVMP class PII—Toxin |
| Test 13 | Toxin | SVMP class PIII—Toxin |
| Test 14 | Toxin | SVSP—Toxin |
| Test 15 | Potential toxin | VEGF—Toxin |

more appropriate measurements of performance than simple accuracy. Another issue of toxin classification lies in unbalanced datasets, because most venomous animal genomes encode less than 100 toxins and 20,000–30,000 physiological non-toxic proteins; as a result, even a high performing method can generate a high number of false positive predictions. For example, 99.5% correct prediction of non-toxins results in ~100 false positive toxins for an average animal proteome, which is in fact more than the actual number of true toxins. In order to minimize both of these problems, the ToxClassifier training scheme was conservative, using only well-annotated toxins from UniProtKB/ToxProt database as positives, and while this might lead to a somewhat lower positive prediction rate (due to missing likely toxins which are not annotated as such), it does serve to minimise the false positive rate.

Notably, predictions were less accurate on some genome datasets, especially _Conus_ snail proteins, with low performance metrics observed for all tested annotation methods. This discrepancy was likely caused by the assumption that sequences deposited in the UniProtKB/SwissProt-ToxProt sequence are _bona fide_ toxins, while sequences in the UniProtKB/SwissProt and TrEMBL databases without 'toxin' or 'venom' keywords are not toxins. Given that toxin activity is attributed to most sequences without biological validation, it is likely that the training datasets almost certainly excluded a number of toxin sequences and included some yet unknown toxins as non-toxic. Another limitation of the 'ToxClassifier' lies in the inherent bias of the training sets; an underrepresentation in sequences from certain animal lineages, particularly the basal Metazoa, e.g., Cnidaria, could lead to incorrect assignment and suspicious quality of existing annotation of conotoxins is a reason to treat prediction on this protein class with caution. To elevate these problems, 'ToxClassifier' has been designed to also report sequences suspected to

have closest homology to underrepresented taxa as 'suspicious toxin' and recommends manual annotation with other tools, such as InterProScan (*Zdobnov & Apweiler, 2001*).

Use of machine learning for toxin prediction has been attempted before and a range of such tools exists; however, most of the available tools are heavily specialised for toxins of specific animal origins. For example, SpiderP (*Wong et al., 2013*) (http://www.arachnoserver.org/spiderP.html) is a predictor for spider toxins while ToxinPred (*Gupta et al., 2013*) (http://crdd.osdd.net/raghava/toxinpred/) predicts only small peptide toxins; while ClanTox (*Kaplan, Morpurgo & Linial, 2007*) (http://www.clantox.cs.huji.ac.il/tech.php) was trained only on an ion-channel toxin dataset and PredCSF (*Fan et al., 2016*) (http://www.csbio.sjtu.edu.cn/bioinf/PredCSF/) is conotoxin specific. In addition, the reported training set sizes are low (for example ClanTox was trained on ∼600 ion channel toxins; the ToxinPred toxin positive training set is 1,805 sequences, while as of 11th May 2016, the UniProtKB/SwissProt-ToxProt database contained ∼6,500 sequences). None of the currently available machine learning methods also gives a comparison with other currently used accepted bioinformatics annotation methods. When compared to ToxClassifier and conventional annotation tools (Tables 2 and 3), ClanTox and ToxinPred tools were found to perform similar to BLAST based methods, while ToxClassifier demonstrated higher performance across all metrics, which is likely a result of comparatively larger training sets and combination of different internal classifiers.

In addition to high performance, the user interface of the 'ToxClassifier' web service reports the best-scoring hit annotation either to UniProtKB/SwissProt-ToxProt (allowing placement of the toxin into the most appropriate toxin protein family), or to the best hit in UniProtKB/SwissProt (giving the closest homology to a non-toxin protein). In summary, this study has established baseline prediction accuracies for a selection of toxin annotation methods and integrates these methods into an easy-to-use, high-precision, machine learning-based classification system named 'ToxClassifier.' This tool offers a reliable and reproducible framework for toxin annotation to enable standardised toxin prediction in venomics projects and to allow for semi-automatic annotation or re-annotation of existing datasets.

## ACKNOWLEDGEMENTS

The authors are grateful to Prof. Dr. Ana M. Moura-da-Silva who provided the unpublished anonymised snake venom transcriptome sequences. We also thank Prof. J Malcolm Shick and Prof. Dr. John Cullum for critically reviewing the manuscript.

### Funding

This work was supported by the United Kingdom Medical Research Council (MRC grant G82144A). PFL is also supported as a Visiting International Research Professor by the Universidade de São Paulo (USP grant 13.1.1502.9.8). The funders had no role in study design, data collection and analysis, decision to publish, or preparation of the manuscript.

## Grant Disclosures

The following grant information was disclosed by the authors:

United Kingdom Medical Research Council: G82144A.

Universidade de São Paulo: 13.1.1502.9.8.

## Competing Interests

The authors declare there are no competing interests.

## Author Contributions

- Ranko Gacesa performed the experiments, analyzed the data, wrote the paper, prepared figures and/or tables, performed the computation work.
- David J. Barlow analyzed the data, reviewed drafts of the paper.
- Paul F. Long conceived and designed the experiments, analyzed the data, wrote the paper, prepared figures and/or tables.

## Data Availability

The raw data has been supplied as a Supplementary File.

## Supplemental Information

Supplemental information for this article can be found online at http://dx.doi.org/10.7717/peerj-cs.90#supplemental-information.

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
