# Peer review of "Machine learning can differentiate venom toxins from other proteins having non-toxic physiological functions"

_PeerJ Computer Science, doi:10.7717/peerj-cs.90_

## Round 0.1 · original submission · Major Revisions

Please thoroughly address all comments from the reviewers. Both reviewers agree that the experimental design needs more detail to really assess its validity e.g. in relation to hyper-parameter tuning and improving the comparison with other methods.

Please explain why from all the methods described in the related work only two of them were included in the quantitative comparison.

·

Basic reporting

The authors present here a meta-approach that combines 9 different machine learning tools for the classification of toxic proteins from their non-toxic counterparts.
The text is well written and the rationale behind the method is expressed in a clear and concise manner.

Experimental design

Results:
In line 111, the authors state that "any blastp hits with duplicated FASTA sequences in the Uniprot/SwissProt-ToxProt database were discarded". Do the authors allow for any variability or do they refer to identical (100%) sequences here? Similarly, the authors should indicate how they handle internal homology in the training sets. The manuscript lacks a detailed description of the creation of the k-folds. My concern is that the authors might be incurring in overtraining if similar (i.e. non-identical globally but closely identical in the tox-bits) signals stemming from closely related proteins are considered in the training and the evaluation sets. And of course, it hinders reproducibility.

The threshold for the selection of valid results is decided after the training, and therefore, possibly introducing overfitting. The authors correctly incorporate a leave-one-out approach in the 25-fold crossvalidation and separate the dataset into training and evaluation. That would be a valid approach if there was no parameter selection in the assessment. As the threshold (i.e <3 non-toxic, 4-5 pot. toxic, 6> toxic) is used afterwards, the distribution of the cross-validation should include three sets: training, test and evaluation. A similar reasoning applies to the selection of the 9 internal methods for classification, as feature selection occurred after the training.

Although the authors correctly include commonly used statistics for the evaluation of machine learning approaches in the Supplementary Note 1B, the numbers for these are not provided and the results are discussed mainly in terms of accuracy. In the problem at hand it is interesting to now the accuracy of prediction on the positive set but also the ability to recall sequences.

Similarly, Figure 1 should be reconsidered. Probably providing together specificity and sensitivity (i.e. a ROC curve). The legend could still be used to display the performance for each cut-off valies and the values for the naiveBLAST and oneBLAST methods. From the observation of the figure, it is not clear to me how the performance was calculated in the manuscript. Again, the all statistics measuring performance are needed. In particular, those focusing on the positive (i.e. toxic proteins) part: precision, recall, f-score and MCC to cite some examples. From the observation of the figure, it seems that the authors present a method very good at predicting the negative class (non-toxic, majority class) but with low recall in the positive class (toxic).

In general, comparisons in the manuscript should be done with respect to precision (accuracy on the positive class) rather than overall accuracy as the prediction of non-toxic proteins is not relevant in this case and constitutes a big majority of the training set.

The same applies to Table 1. Please, report all statistics for all the methods to facilitate comparison. Or even include them in Figure 1 for direct comparison.

In Table 3, please, provide the individual scores.

Validity of the findings

In the paragraph starting in line 246 the authors describe other approaches that, although limited in scope, address a similar problem. However, the performance of these specific method is not critically compared to the performance of the ToxClassifier presented here. The readership of the manuscript would definitely benefit from a detailed benchmark with these in addition to the other sequence based generic approaches.

Additional comments

The Methods section needs improvement to reach publication standards. I would suggest that the information in the Supplementary Notes is expanded and included into the Methods section. In particular, the sections referring to the different annotation systems and the statistics used for the evaluation of the methods constitute a core part of the interpretation of the results and should be described carefully in the methods sections. I mentioned above some problems with respect to the generation of the training and evaluation sets in the k-fold crossvalidation approach used. This should constitute a section in the Methods as well.

In line 49, the sentence "Thus, remote .... approach alone" might need rewording. The meaning is not clear to me in its current form.

Reviewer 2 ·

Basic reporting

The paper is well written and the aims are clearly stated. The introduction and background provided are adequate. Figures are relevant and the explanation that is included on them is enough to understand their content.

The literature included is relevant and up-to-date.

Experimental design

The paper presents the results of the use of standard machine learning classification methods to distinguish between toxic and non-toxic peptides and proteins.

The experiments are well designed and the databases are constructed in a principled way. However, there is a lack of comparison with other relevant programs aimed at the same goal which are described in the paper but not used as a baseline to evaluate the validity of the proposed approach.

From the computational point of view not sufficient information is given about how the models were trained or how the hyper-parameters of the models where chosen. This would make the reproduction to the experiments troublesome.

Validity of the findings

Although the experiments and results are valuable, I think the novelty of the paper is weak. The application of standard machine learning methods to toxic/non toxic protein classification is interesting but it is not a clear innovation over other existing programs that use other similar techniques.

I think that the author should justify the contribution of their paper to the field in addition to the seemingly good results achieved.

---

## Round 0.2 · accepted · Accept

The authors have thoroughly addressed the concerns of the reviewers and the paper is now ready for publication.

·

Basic reporting

The authors have carefully addressed my concerns to the previous version of the manuscript. Therefore, I have no more objections to its publication.

Experimental design

No Comments

Validity of the findings

No Comments

Additional comments

No Comments